# Physiological and Transcriptomic Analysis Revealed the Molecular Mechanism of *Pinus koraiensis* Responses to Light

**DOI:** 10.3390/ijms232113608

**Published:** 2022-11-06

**Authors:** Yuxi Li, Xinxin Zhang, Yan Zhu, Kewei Cai, Hanxi Li, Qiushuang Zhao, Qinhui Zhang, Luping Jiang, Yan Li, Tingbo Jiang, Xiyang Zhao

**Affiliations:** 1State Key Laboratory of Tree Genetics and Breeding, Northeast Forestry University, Harbin 150040, China; 2College of Forestry and Grassland, Jilin Agricultural University, Changchun 130118, China

**Keywords:** *Pinus koraiensis*, supplementary light, physiological, transcriptomic, circadian rhythm-plant

## Abstract

Korean pine (*Pinus koraiensis* Sieb. et Zucc.), as the main tree species in northeast China, has important economic and ecological values. Currently, supplementary light has been widely used in plant cultivation projects. However, the studies about different supplementary light sources on the growth and development of Korean pine are few. In this study, the one with no supplementary light was used as the control, and two kinds of light sources were set up: light-emitting diode (LED) and incandescent lamp, to supplement light treatment of Korean pine. The spectrum and intensity of these two light sources were different. The results showed that the growth and physiological–biochemical indicators were significantly different under different supplementary light treatments. The biomass of supplementary light treatment was significantly lower than the control. Compared with the control, IAA and GA were lower, and JA, ABA, ZT, and ETH were higher under supplementary light conditions. Photosynthetic parameters in supplementary light conditions were significantly lower than the control. Supplemental light induces chlorophyll a, chlorophyll b, total chlorophyll, and carotenoid accumulation. From RNA-seq data, differentially expressed genes (DEGs) were observed in all the comparison groups, and there were 487 common DEGs. The expression levels of DEGs encoding transcription factors were also changed. According to GO and KEGG analysis, the plant hormone signal transduction, circadian rhythm-plant, and flavonoid biosynthesis pathways were the most enriched. These results provided a theoretical basis for the response of Korean pine to different supplementary lights.

## 1. Introduction

Korean pine (*Pinus koraiensis* Sieb. et Zucc.), belonging to the *Pinus* genus and the Pinaceae family, gymnosperms group, is a major five-leaved pine, and is the dominant tree species in the northern hemisphere [1]. It is mainly located in the far eastern region of Russia, northeast China, North and South Korea, and some parts of Japan [2]. Korean pine is the pioneer tree species in mixed coniferous broad-leaved forests, which has important ecological value. Moreover, Korean pine has been listed as a national second-class protected plant as early as 1999 [3]. Timber of Korean pine has excellent characteristics such as lightness, softness, corrosion resistance, and easy processing. Therefore, it is often used in the manufacture of furniture, the construction of ships and bridges, and so on. At the same time, Korean pine nuts have high edible value, because they are rich in unsaturated fatty acid, amino acids, etc. [4]. Due to the various bioactive substances they contain, such as polysaccharides, flavonoids, and phenolic compounds, they have antitumor, antibacterial, and antioxidant activities [5].

Light is the paramount energy source in plant growth and development progress, including seed germinating, chlorophyll accumulation, flower induction, and circadian rhythms [6,7]. At present, supplementary light has been widely used in agricultural production. Among various supplementary light sources, light-emitting diode (LED) has become the most commonly used because of its numerous advantages [8]. LED, as a cold light source, has a relatively cool emitting surface, as well as small size, durability, and a long operating lifetime [9]. Supplementary lights with different light qualities also affect plant biomass accumulation, pigment synthesis, and antioxidants [10,11]. Fukuda et al. used red, blue, and white lights provided by LED to illuminate *Petunia hybrid*, and the study showed that red light inhibited shoot elongation, while blue light greatly increased it [12]. However, other studies have shown that blue light also suppressed stem elongation [13]. On the other hand, chlorophyll a and chlorophyll b could effectively absorb blue and red lights [14]. In addition, not only light quality, but also circadian rhythm greatly affected flowering [15].

Higher plants can sense changes in the light environment, including light quality, quantity, direction, and duration. To sense and respond to these fluctuations in the light environment, higher plants possess several families of photoreceptors [7], which mainly contain phytochromes, cryptochromes, phototropins, and UVR8 [16]. The photoreceptors of plants can respond to different wavelengths in the spectrum. Phytochromes (phyA–phyE) maximally absorb in the red/far-red region of the light spectrum [17]. Cryptochromes (cry1 and cry2) and phototropins (phot1 and phot2) absorb principally in the blue/UV-A region [7,18]. UVR8 has been known to be mainly responsible for sensing and absorbing ultraviolet light [19]. The signaling pathways of the photoreceptors described above are integrated to adapt to changing light environments [20]. In addition, a proper functional circadian clock is essential for plant survival [6]. It is involved in numerous biological processes, including internal metabolic and hormonal signals, etc. [21,22]. Plants with impaired clocks show dramatic changes in flowering time and viability [6,23].

At present, a majority of studies on *P. koraiensis* mainly focus on growth traits, wood characteristics, genetic improvement, and so on [24,25,26]. Several previous studies have demonstrated that pine seedlings have different growth and physiological responses to different LED spectrums [27,28]. Wei et al. studied the quality estimation, nutrient concentration, foliar physiology, and enzyme activity of Korean pine seedlings under different spectral conditions [29]. However, little is known about the molecular mechanism of the supplemental light of Korean pine under different light spectrums. In this study, different supplementary light sources were set up, and the responses of Korean pine to different light spectrums were comprehensively explored by measuring the biomass, hormones, photosynthesis, and pigments, as well as transcriptome sequencing.

## 2. Results

### 2.1. Effect of Supplementary Light on Biomass

There were significant differences (*p* < 0.01) in the fresh weight and dry weight of shoots and roots (Appendix A). Compared with the CK (control), fresh weights of shoots and roots under two supplementary light treatments were both significantly lower (Figure 1A). As shown in Figure 1B, dry weights of shoots and roots of the CK were both significantly higher than those with the supplementary light treatments; moreover, the dry weight of shoots under L4 (supplementary LED for 4 h) treatment was significantly higher than those under In4 (supplementary incandescent lamp for 4 h) treatment.

### 2.2. Effect of Supplementary Light on Physiological–Biochemical Indexes

The concentrations of auxin (IAA), gibberellin (GA), ethylene (ETH), abscisic acid (ABA), jasmonic acid (JA), salicylic acid (SA), brassinosteroid (Br), and zeatin (ZT) were significantly differentially expressed (*p* < 0.01) in all treatments (Appendix A). In addition, IAA and GA concentrations of the CK were significantly higher than in the two supplementary light treatments (Figure 2A,B). On the other hand, the levels of ETH, ABA, JA, SA, Br, and ZT in the CK treatment were significantly lower than those in the supplementary light treatments (Figure 2C–H). Furthermore, the IAA concentration of In4 treatment had the lowest level among all treatments, but ABA and Br were significantly higher under the In4 treatment than those in the other treatments. In the L4 treatment, the contents of JA, SA, ZT, and ETH were the highest among all treatments.

For photosynthesis analysis, all photosynthetic parameters showed significant differences (*p* < 0.01) in all treatments (Appendix A). As shown in Figure 3A–E, all supplementary light treatments induced a significant decrease in the net photosynthetic rate (Pn), stomatal conductance (Gs), intercellular CO_2_ concentration (Ci), transpiration rate (Tr), and water use efficiency (WUE) compared with the CK treatment. Gs and Tr in L4 were significantly lower than those of the other treatments. Furthermore, Ci and WUE in In4 were the lowest among all treatments.

For the total chlorophyll, chlorophyll a (Chla), chlorophyll b (Chlb), and carotenoid concentrations in different supplementary light treatments, there were significant differences (*p* < 0.01) among all treatments (Appendix A). The results showed that the supplementary light treatments significantly increased the content of total Chl, Chla, Chlb, and carotenoids (Figure 3F–I).

### 2.3. Transcriptomic Analysis

A total of nine cDNA libraries were obtained from the three treatments (three replicates for each treatment), and they were sequenced on the Illumina platform. Then, 76.70 Gb of clean base was collected from the 9 samples, with more than 6.00 Gb of clean base obtained from each sample. The fragments scoring Q20 and Q30 of every sample were both more than 98.10% and 94.20%, respectively (Appendix A). The average GC content of all samples ranged from 42.97% to 46.05%. A total of 540,142,160 raw reads were generated from the 9 cDNA libraries. Finally, 511,304,616 clean reads were obtained after filtering out the ambiguous nucleotides and low-quality reads. De novo transcriptome was assembled by Trinity [30] to obtain 169,624 unigenes, with N50 length of 1374 bp. The clean reads were aligned to the assembled sequences by RSEM [31], and an average of 85.75% clean reads were mapped to the assembled sequences. The assembly quality was evaluated by using BUSCO software, and the results showed that 1614 genes were detected with a coverage rate of 91.90% (1482 genes) (Appendix A). These results indicated a high-quality assembly and a high level of completeness.

Gene function annotation was carried out, and the unigene sequences were mapped to public databases by using DIAMOND [32] BLASTX, with the following results. A total of 79,318 (46.76%) unigenes matched to a sequence in at least one of the following databases. The number of unigenes was 75,329 (44.41%) in the Nr (nonredundant protein sequences) database, followed by 74,197 in the TrEMBL (43.74%), 63,525 in the GO (Gene Ontology) (37.45%), 51,949 in the KEGG (Koyoto Encyclopedia of Genes and Genomes) (30.63%), 51,020 in the Swiss-Prot (a manually annotated and reviewed protein sequence database) (30.08%), 50,187 in the Pfam (Protein family) (29.59%), and 41,321 in the KOG (eukaryotic Orthologous Groups) (24.36%) databases. According to a search on the GO database, a total of 59 terms were concentrated: the number of ‘biological process’ was the largest, which was 28, followed by 18 in the ‘cellular component’ and 13 in the ‘molecular function’ (Figure 4) (Appendix A). By alignment of unigene sequences with the Nr database, the unigene sequences had the strongest matches with the gene sequences from *Picea sitchensis* (38.85%), followed by *Quercus suber* (6.72%), *Amborella trichopoda* (5.45%), *Nelumbo nucifera* (2.99%), *Picea glauca* (2.97%), *Pinus taeda* (2.49%), *Pinus tabuliformis* (1.78%), *Macleaya cordata* (1.45%), *Marchantia polymorpha* subsp. ruderalis (1.35%), *Physcomitrella patens* (1.32%), *Vitis vinifera* (1.32%), and *Pinus radiata* (0.99%) (Figure 5). The unigene sequences matched to the KOG database according to the sequence similarity, and the results showed that the top three classes were: (R) ‘general function prediction only,’ (O) ‘post-translational modification, protein turnover, chaperones,’ and (T) ‘signal transduction mechanisms’ (Figure 6).

### 2.4. Annotation and Enrichment Analysis of the Differentially Expressed Genes (DEGs)

Firstly, the results of PCA (principal component analysis) showed that these samples revealed effective separation (Figure 7A). The DEGs were observed in all the comparison groups (Figure 7B). There were more DEGs in the CK vs. L4 group, which contains 6428 DEGs, with 2349 upregulated and 4079 downregulated. In the CK vs. In4 comparison group, there were 5206 DEGs (2233 upregulated and 2973 downregulated). In addition, there were 6068 DEGs (3770 upregulated and 2298 downregulated) in the L4 vs. In4 comparison group. The Venn diagram showed that there was a total of 487 common DEGs in the three comparison groups (Figure 7C).

To perform further functional analysis of DEGs, GO annotation and KEGG enrichment analysis were performed. The results of GO functional classification showed that there was a total of 50, 51, and 49 terms in CK vs. L4, CK vs. In4, and L4 vs. In4 comparison groups, and they were divided into “biological process”, “cellular component”, and “molecular function” categories (Appendix AA–C). In the three comparison groups, “biological process” was the most enriched classification, which mainly included metabolic process, cellular process, and response to stimulus. At the same time, the terms cell, cell part, and organelle were the most enriched in the “cellular component” category. On the other hand, the terms associated with “molecular function” were the least enriched, which mainly included catalytic activity, binding, and transporter activity.

The top 20 KEGG enriched pathways of the DEGs from the three comparison groups are shown in Figure 7D–F. There were 133 pathways enriched in the CK vs. L4 comparison group, and therein, flavonoid biosynthesis, brassinosteroid biosynthesis, and metabolic pathways were the most enriched KEGG pathways (Figure 7D) (Appendix A). In the CK vs. In4 comparison group, the most enriched KEGG pathways were cutin, suberine, and wax biosynthesis, flavonoid biosynthesis, and fatty acid elongation (Figure 7E) (Appendix A). Flavonoid biosynthesis, flavone and flavonol biosynthesis, and metabolic pathways were the most enriched KEGG pathways in the L4 vs. In4 comparison group (Figure 7F) (Appendix A). It can be seen that the flavonoid biosynthesis pathway was enriched in all comparison groups. At the same time, plant hormone signal transduction, carotenoid biosynthesis, and circadian rhythm-plant were all among the top 20 enriched pathways in the three comparison groups. These pathways may play a key role in the process of light supplementation in *P. koraiensis*.

### 2.5. Analysis of Transcription Factors (TFs)

TFs play a key role in the growth and development of plant progress. In this study, a total of 415 DEGs were identified as TFs. They mainly included MYB-related (13%), AP2/ERF (12%), MYB (7%), Tify (6%), NAC (5%), WRKY (5%), bHLH (5%), HB (5%), C2H2 (4%), bZIP (3%), LOB (3%), and other TFs (32%) (Figure 8A) (Appendix A). As shown in Figure 8B–E, MYB-related, AP2/ERF, MYB, and Tify were all significantly expressed in different supplementary light treatments (Appendix A). The majority of DEGs of the top three TF families (MYB-related, AP2/ERF, and MYB) were downregulated in the In4 treatment, and notably, all DEGs of the Tify family were downregulated in the In4 treatment. Interestingly, the majority of DEGs of the top four TF families above were upregulated in the L4 treatment.

### 2.6. Effect of Different Supplementary Light Sources on Plant Hormone Signal Pathway

For plant hormone signal transduction pathway analysis, DEGs were enriched in eight hormone signal transduction pathways (Figure 9). The results showed that there was a total of 191 genes encoding 30 enzymes (Appendix A). In the auxin signal transduction pathway, all AUX1, AUX1/IAA, and GH3 genes were upregulated in L4 and downregulated in In4. All ARF genes (except Cluster-27241.81122) were upregulated in both L4 and In4. The majority of genes encoding SAUR were upregulated in L4 and downregulated in In4. For zeatin signal transduction pathway analysis, gene CRE1 was downregulated in the two supplementary light treatments, and the majority of genes encoding B-ARR were upregulated in L4 and downregulated in In4. In the gibberellin signal transduction pathway, the majority of genes encoding GID2, DELLA, and TF were upregulated in the two supplementary light treatments. The majority of PYR/PYL genes in the abscisic acid signal transduction pathway were downregulated in L4 and In4, however all ABF and PP2C genes were upregulated in L4 and In4. There were nine DEGs encoding five enzymes in the ethylene signal transduction pathway. In the brassinosteroid signal transduction pathway, there were 51 DEGs encoding 5 enzymes. The majority of BRI1, BSK, and TCH4 genes were upregulated in L4, and downregulated in In4. However, most BAK1 and BKI1 genes showed the opposite trend. In the jasmonic acid signal transduction pathway, DEG COI1 was downregulated and all MYC2 genes (except Cluster-27241.104920) were upregulated in the two supplementary light treatments. A total of nine genes encoding PR-1 were identified as DEGs in the salicylic acid signal transduction pathway, and all DEGs (except Cluster-27241.33366 and Cluster-27241.78060) were upregulated in L4.

### 2.7. Effect of Different Supplementary Light Sources on Circadian Rhythm-Plant Pathway

From the results of KEGG enriched pathways in this study, it can be seen that circadian rhythm-plant was one of the top 20 enriched pathways in all comparison groups. There were 53 genes encoding 10 enzymes that were significantly differentially expressed in different treatments (Figure 10) (Appendix A). In the two supplementary light treatments, PHYA, CRY, and CO genes were upregulated and the COP1 gene was downregulated. The majority of PHYB genes were also upregulated in the two supplementary light treatments. All DEGs encoding LHY (except Cluster-27241.45170) were upregulated in the two supplementary light treatments compared with the CK. All CK2α DEGs were upregulated in In4, and all CK2β DEGs were upregulated in L4. Meanwhile, most genes encoding CHS were upregulated in L4, and were downregulated in In4.

### 2.8. Effect of Different Supplementary Light Sources on Flavonoid Biosynthesis

There was a total of 107 genes encoding 9 enzymes in flavonoid biosynthesis (Figure 11) (Appendix A). The majority of CHS DEGs were upregulated in L4, and downregulated in In4, however, only Cluster-27241.71072 showed the opposite trend. Interestingly, the results of the expression levels of F3H, FLS, F3′5′H, F3′H, and DFR showed that they had similar trends to CHS. However, most DEGs encoding ANS were downregulated in In4. In addition, all LAR genes were upregulated in L4.

### 2.9. Quantitative Real-Time PCR (qRT-PCR) Validation

To validate the transcriptome data, a total of nine DEGs were randomly selected for expression analysis in the three samples by using qRT-PCR. As shown in Appendix A, these DEGs displayed similar expression trends in RNA-seq and qRT-PCR. These results indicate the reliability of Illumina sequencing.

## 3. Discussion

To reveal the mechanism of the responses of *P. koraiensis* to different supplementary light sources, growth and physiological indicators, as well as transcriptome sequencing, were performed (Figure 12). In this study, the biomass in *P. koraiensis* under supplementary light at night for 4 h was lower than that in the control. Li et al. confirmed that proper supplementary light can increase the biomass of baby leaf lettuce [33], and a similar result was obtained in the study of cucumber by Hao et al. [34]. However, the result of this study is different from these studies. Several studies have shown that tomato grew and yielded optimally with a photoperiod of 14 h of light. The yield of tomato did not increase after a longer light period, and instead, 20 or 24 h of light even reduced the yield [35,36]. This experimental site has a sunlight time from sunrise to sunset of up to 16 h in June and July, then supplementary light for 4 h to make the lighting time reach 20 h. Therefore, this paper inferred that too-long lighting duration will adversely affect the growth of *P. koraiensis*. It is well-known that sugar derived from photosynthesis is the main source of carbon and energy in plants [37]. In this study, all the photosynthetic parameters were significantly decreased under supplementary light compared with the control. At the same time, the shoot dry weight and Ci of In4 were the lowest among all treatments. The shoot dry weight in In4 was the lowest possibly due to the change in photosynthetic carbon fixation caused by the decrease in Ci. As shown in Figure 1, the spectrum of incandescent lamps lacked blue light, while red light was more dominant. Blue light generally promotes stomata opening more than other light wavelengths, which may lead to a higher photosynthetic capacity and biomass in plants under blue light [38,39,40]. Sander et al. showed that the photosynthetic rate of rice under red light alone was lower than that of a mix of red and blue lights [41]. Excessive red light and a lack of blue light could affect photosynthesis and carbon fixation in plants. In addition, the intensity of the incandescent lamp was much greater than LED in this study, and the incandescent lamp generated more heat than LED (Figure 1). Excessive light could cause photooxidative damage and photoinhibition [42]. Under strong illumination, a series of changes at the acceptor side PSⅡ led to the inhibition of electron transport, which ultimately affected the photosynthesis [43]. This reduces the net photosynthetic rate of Korean pine under supplementary light conditions, while the chlorophyll content increases. At low photosynthetic rates, plants increase their chlorophyll content to improve photosynthetic rates. However, the photoinhibition of Korean pine caused by excessive light causes the net photosynthetic rate to decrease.

Changes in various exogenous growth conditions modulated a panel of endogenous plant hormones. In turn, these hormones ensured maximum fitness throughout the plant lifecycle by regulating a range of physiological processes [44]. Responses of plants to changes in the light environment were directly regulated by hormones or by hormonally regulated metabolic changes [45]. In general, plant hormones interact with each other. For example, the change of the light signal will change the content of auxin, cytokinin, ABA, GA, and BR [44,46,47,48]. In this study, phytohormones IAA, GA, JA, SA, ABA, ZT, ETH, and BR all changed under supplementary light of different light sources, and the differences were significant. At the same time, the genes of the eight hormone signaling transduction pathways were induced. This suggests that the hormones of *P. koraiensis* respond to different spectrums of supplementary light. Many phytohormones are known to control specific features of the plant-circadian system in distinct ways [49], and the regulation of all aspects in phytohormones is under the clock control. However, the systematic description referring to the integration of hormonal signals into the plant-circadian system is currently lacking. It is suggested that this study can provide some new progress for this aspect of research.

The circadian rhythm is widespread in plants, and the transcriptional circadian cycles have been reported in a wide variety of species [50]. The circadian clock is an endogenous time-keeping mechanism that requires approximately 24 h to complete a signal cycle [51]. In this study, supplementary lighting at night extended the duration of light, and therefore changed the photoperiod of the Korean pine. At the same time, the genes related to the circadian rhythm-plant pathway were induced under different supplementary light sources. LHY genes in this study were upregulated in supplementary lighting treatments, and this may be due to LHY responding to changes in light duration. CCA1 and LHY are reported in lighting regulation in the plants [52]. Furthermore, a large number of genes encoding CHS were differentially expressed under different treatments and showed similar trends. Currently, few reports have found that CHS genes play an important role in the circadian rhythm-plant pathway. However, CHS may play a key role in the regulation of *P. koraiensis* under different supplementary light conditions. In the circadian systems, the central oscillator consists of three interlocking, auto-regulatory transcriptional feedback loops [53]. It is a complex system that requires more in-depth study by researchers. In addition, Wei et al. found that 16 h of lighting per day significantly increased the H_2_O_2_ content and antioxidant enzyme activities of seedlings by treatments with different durations of lighting in watermelon. These results indicated that 16 h of supplementary light led to certain stresses in watermelon seedlings [54]. The circadian system is a complex, interconnected, and reciprocally regulated network, and the correct timing of circadian rhythms conferred an adaptive advantage [6,55]. This study analyzed the data of flavone metabolism and the result showed that numerous genes were induced in different supplementary light treatments, where a majority of DEGs were upregulated in L4 and downregulated in In4. A large number of studies have demonstrated that flavonoids can respond to changes in the light environment. The study of Giliberto et al. has shown that blue light increased levels of anthocyanins in tomato [56]. In this study, there was more blue light in L4 than in In4. Therefore, the DEGs related to the flavonoid pathway were induced in L4. The study by Li et al. has proven that *P. koraiensis* could accumulate a large amount of flavonoids in a light stress environment [57]. These results suggested that *P. koraiensis* does not need too much light during the growth and development progress.

## 4. Materials and Methods

### 4.1. Plant Materials and Treatments

Korean pine is mainly located in the region of northeast China. The experiment was conducted in Daquanzi Forest Farm in Harbin City, Heilongjiang Province (126°55′–128°19′, 45°30′–46°01′). The experimental materials consisted of four-year-old Korean pine seedlings that were grown in the local region. There were two supplementary illumination treatments, including white LED and incandescent lamp lighting, and four hours of different supplementary light source treatments were applied after sunset every day. The one with no supplementary light was treated as the control (CK), L4 was supplied with 4 h of LED supplementary light, and In4 was supplied with 4 h of incandescent lamp supplementary light. The LED and incandescent lamp were provided by Shanghai Yaming Illumination Co., LTD. The spectral qualities of the white LED (Appendix A) and incandescent lamp (Appendix A) were measured by a spectrometer (Ocean insight HR4000, Dunedin, FL, USA), and the results are shown in Figure 13. The peaks of the white LED were 451 and 563 nm, and the peak of the incandescent lamp was 655 nm. All light sources were fixed to 0.5 m above the top leaf of the seedlings. There was a total of 50 seedlings per treatment, and blackout cloth was used for isolation between different treatments. To avoid the influence among different light sources, the sunlight was not blocked during the day. The experiment lasted from 11 June to 11 September 2021, and routine management practices were carried out during this period.

### 4.2. Measurement of Seedling Biomass

After supplementary light treatment, three seedlings were randomly selected to measure the biomass. The entire *P. koraiensis* seedlings were carefully removed from the soil without damaging the roots. Then, they were rinsed in the distilled water and wiped clean. The seedlings were divided into aboveground and underground parts and weighed separately for their fresh weight. Dry weight was determined by drying at 80 °C until constant weight after heating at 105 °C for 20 min.

### 4.3. Measurement of Plant Hormones

Enzyme-linked immunosorbent assays (ELISA) were used to measure IAA, GA, ETH, ABA, SA, JA, Br, and ZT contents. Three *P. koraiensis* seedlings were randomly selected from each treatment, and the leaves at the same position were selected for mixed sampling, which were immediately frozen in liquid nitrogen to measure the phytohormones. There were three biological replicates in each treatment. The hormone extraction method refers to the method of Zhang et al., with minor modifications [58]. After grinding the sample in liquid nitrogen, 0.5 g of powder was weighed and then 2 mL of 80% methanol was added into it. It was stored at 4 °C for 16 h to extract hormones, then the supernatant was absorbed into a new centrifuge tube after centrifugation (5000× *g* at 4 °C, 10 min). The residues were re-extracted with 80% methanol according to the methods above and all the supernatants were merged together. Finally, the liquid was dried by using nitrogen gas and redissolved with methanol for ELISA analysis. The ELISA kit was supplied by Shanghai Enzymatic Biotechnology Company Ltd. (Shanghai, China). All specific steps were performed according to the manufacturer’s instructions.

### 4.4. Measurement of Photosynthetic Parameters and Photosynthetic Pigments

Photosynthetic parameters were measured by using the portable CIRAS-3 photosynthesis system (PP Systems, Amesbury, MA, USA). Photosynthetic parameters included Pn, Gs, Ci, Tr, and WUE. Measurement of photosynthesis was carried out between 9:00 and 11:00 a.m. on sunny days. Light was provided during measurement, with natural light in real time, and CO_2_ concentration was that of the atmospheric air. Other parameters were not controlled. For each Korean pine seedling, needle leaves of the same height and the same direction were chosen to measure. Three seedlings were selected for each treatment and three replicates were measured per seedling.

Photosynthetic pigments were measured according to the study of Zhang et al. with a few modifications [59]. Three seedlings were randomly selected from each treatment, and the leaves in the same position were selected, wiped clean, and mixed. The mixed leaves were immediately frozen in liquid nitrogen for extracted pigments. Mixed leaves were ground in liquid nitrogen, then 10 mL of 95% ethanol was added to 0.1 g of powder. They were put in the dark until blanched, and then the supernatant was absorbed to measure the absorbance values at 649, 664, and 470 nm. The whole experiment was carried out in low light. Finally, chlorophyll a, chlorophyll b, total chlorophyll, and carotenoid concentrations were calculated by using the equation described by Zhang et al. [59]. There were three replicates from each treatment.
$$Chla=13.36\times A_{664}-5.19\times A_{649}\;Chlb=27.43\times A_{649}-8.12\times A_{664}\;Car=\left(1000\times A_{470}-2.13\times Chla-97.64\times Chlb\right)/209\;total\;Chl=Chla+Chlb$$
where *A*_664_, *A*_649_, and *A*_470_ are the measured absorbances at 664, 649, and 470 nm, respectively.

### 4.5. RNA-Seq and de Novo Transcriptome Assembly

Three seedlings were randomly selected for each treatment, and leaves of the same position were collected and mixed, wiped clean, and immediately frozen in liquid nitrogen. The extraction of total RNA from leaves was generated by using the total RNA isolation kit (Tiangen, Beijing, China) following the manufacturer’s instructions. There were three replicates for each treatment. RNA was sent to Wuhan MetWare Biotechnology Co., Ltd. (Wuhan, China) for RNA-seq. The RNA of all samples was quantified by the Nano Photometer spectrophotometer (IMPLEM, Westlake Village, CA, USA), Qubit 2.0 (Life Technologies, Carlsbad, CA, USA), and the Agilent 2100 bioanalyzer (Agilent Technologies, Santa Clara, CA, USA) to estimate the quality, concentration, and integrity. The poly(A) mRNA was isolated from purified total RNA by using oligo (dT) magnetic beads. After purification, fragmentation buffer was added to break the mRNA into short fragments. Then, they were used as the templates and random hexamers to synthesize the first-strand cDNA. Second-strand cDNA was synthesized by using buffer, dNTPs, RNaseH, and DNA polymerase Ⅰ. Double-stranded cDNA was purified by using AMPure XP beads and enriched with PCR to obtain the final library. These libraries were sequenced on the Illumina Hi-Seq^TM^ 2500 (San Diego, CA, USA) with 150 bp lengths. There was a total of nine cDNA libraries from the three samples. To obtain high-quality clean reads, the raw reads were filtered by Fastp software [60]. Firstly, adaptor-containing reads were removed; secondly, the reads containing more than 5% ambiguous nucleotides were removed, and finally, the low-quality reads that contained more than 15% bases with q-value ≤ 19 were removed, and the clean reads were obtained for de novo assembly. De novo transcriptome was assembled by using Trinity [30], and furthermore, these unigenes were spliced to generate longer complete consensus sequences. Then, the unigene sequences were aligned to functional databases, including GO, KOG, KEGG, Nr, Swiss-Prot, Pfam, and Trembl.

### 4.6. Differentially Expressed Genes’ (DEGs) Analysis and Functional Annotation

PCA was performed by using the unsupervised pattern recognition method. The numbers of mapped reads and transcript lengths were normalized in the samples. Then, the gene expression level was measured by FPKM (Fragments Per Kilobase of transcript per Million fragments mapped). Differential expression analysis was conducted with DEseq2 [61,62]. The DEGs were identified according to |log_2_ Fold Change| ≥ 1 and *p*-value < 0.5. For DEGs’ analysis, the GOseq R package was used in GO functional annotation, and KOBAS software was used in the enriched pathways’ analysis of KEGG.

### 4.7. Validation of qRT-PCR

To validate the transcriptome data, nine DEGs were randomly selected to perform the expression analysis in all samples. The RNA was extracted using an RNA isolation kit (Tiangen, Beijing, China). cDNA was synthesized using the Prime Script RT reagent Kit with gDNA Eraser (TaKaRa, Kyoto, Japan). The specific operation steps were carried out according to the product manual. The PCR amplification experiment was performed with the TaKaRa SYBR Green Mix kit (TaKaRa, Kyoto, Japan) (5 μL 2 × SYBR Green premix ExTaq II, 0.2 μL Rox Reference Dye II, 0.8 μL primer-F/R, 1 μL cDNA, and ddH_2_O to 10 μL). Then, qRT-PCR was performed as follows: 95 °C for 30 s, 40 cycles of 95 °C for 5 s and 60 °C for 35 s, 95 °C for 5 s, 60 °C for 1 min, and 95 °C for 15 s. The qRT-PCR tests used the ABI 7500 Fast Real-Time Detection System. All primers are listed in Appendix A, and the reference gene was 18S-RNA. Finally, the relative expression levels were calculated using the 2^−ΔΔCT^ method [63].

### 4.8. Statistical Analysis of Data

The data were analyzed by using Excel 2021 and SPSS version 26.0 (International Business Machines, Armonk, NY, USA). Firstly, to test the effect of all physiological indices, one-way ANOVA was used. The linear model for the analysis of the three treatments was as follows [64]. Then, multiple comparisons of means were conducted by S-N-K tests at *p* < 0.05 to assess the physiological responses of different supplementary light sources.
$$X_{ij}=\mu +T_{i}+e_{ij}$$
where $X_{ij}$ is the performance of an individual tree *j* in treatment *i*, *μ* is the overall mean, $T_{i}\;$is the fixed effect of treatment, and $e_{ij}$ is the random error.

## 5. Conclusions

In this study, the biomass and physiological indexes of *P. koraiensis* were differentially expressed under different supplementary lights. The biomass was lower under supplementary light treatments than in the control. Phytohormones IAA, GA, JA, SA, ABA, ZT, ETH, and BR were all significantly differentially expressed in different supplementary lighting treatments. All photosynthetic parameters under supplementary light were significantly lower than the control. Supplementary light induced chlorophyll and carotenoid accumulations. Furthermore, according to transcriptomic analyses, most of the differentially expressed genes were enriched in plant hormone signal transduction, circadian rhythm-plant, and flavonoids biosynthesis pathways. In summary, long light duration and excessive light can adversely affect the growth of *P. koraiensis*. It can be seen that the growth and development of *P. koraiensis* seedlings do not require too much light.

## Figures and Tables

**Figure 1 ijms-23-13608-f001:**
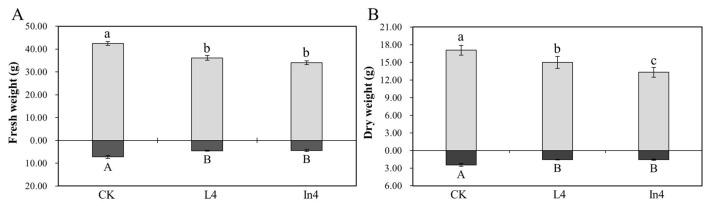
Changes in shoot and root biomass in *P. koraiensis* under different supplementary lights. (**A**) Fresh weight. (**B**) Dry weight. CK: Control. L4: Supplementary LED for 4 h. In4: Supplementary incandescent lamp for 4 h. Letters of a, b, and c are labels for the shoots, and A and B are labels for the roots.

**Figure 2 ijms-23-13608-f002:**
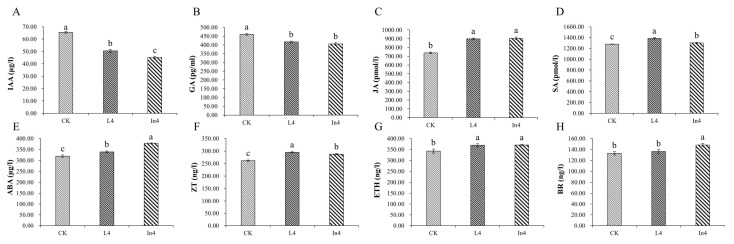
Changes in plant hormone contents in *P. koraiensis* under different supplementary lights. (**A**) IAA, (**B**) GA, (**C**) JA, (**D**) SA, (**E**) ABA, (**F**) ZT, (**G**) ETH, and (**H**) BR. CK: Control. L4: Supplementary LED for 4 h. In4: Supplementary incandescent lamp for 4 h. The error bars represent the standard error. Different letters indicate significant differences among different treatments.

**Figure 3 ijms-23-13608-f003:**
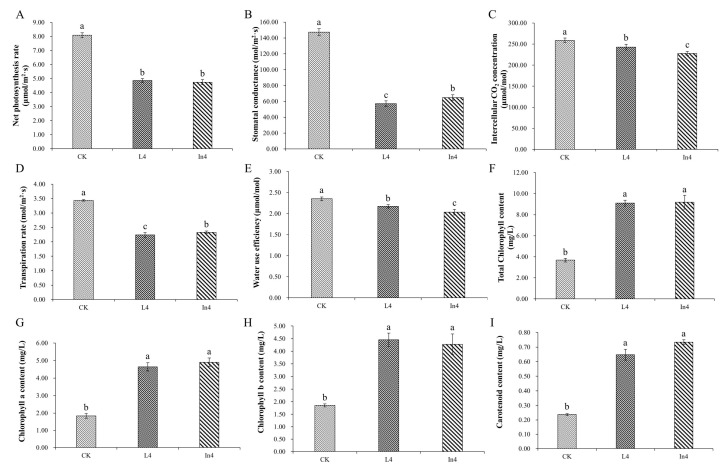
Changes in photosynthetic indicators and pigments in *P. koraiensis* under different supplementary lights. (**A**) Net photosynthetic rate (Pn). (**B**) Stomatal conductance (Gs). (**C**) Intercellular CO_2_ concentration (Ci). (**D**) Transpiration rate (Tr). (**E**) Water use efficiency (WUE). (**F**) Total chlorophyll content. (**G**) Chlorophyll a content. (**H**) Chlorophyll b content. (**I**) Carotenoid content. CK: Control. L4: Supplementary LED for 4 h. In4: Supplementary incandescent lamp for 4 h. The error bars represent the standard error. Different letters indicate significant differences among different treatments.

**Figure 4 ijms-23-13608-f004:**
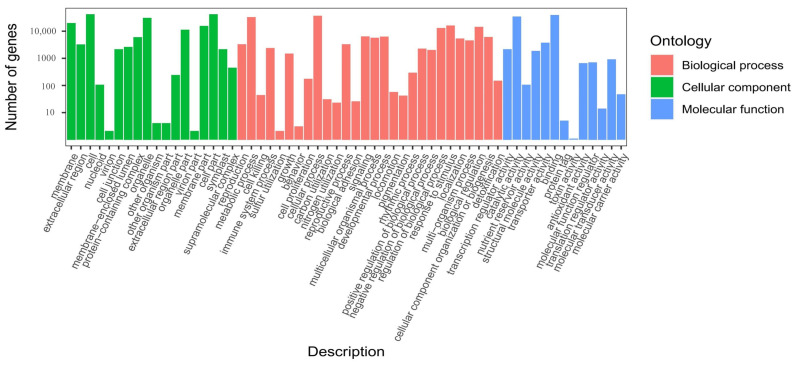
GO classifications of assembly of unigenes in *P. koraiensis* under different supplementary lights.

**Figure 5 ijms-23-13608-f005:**
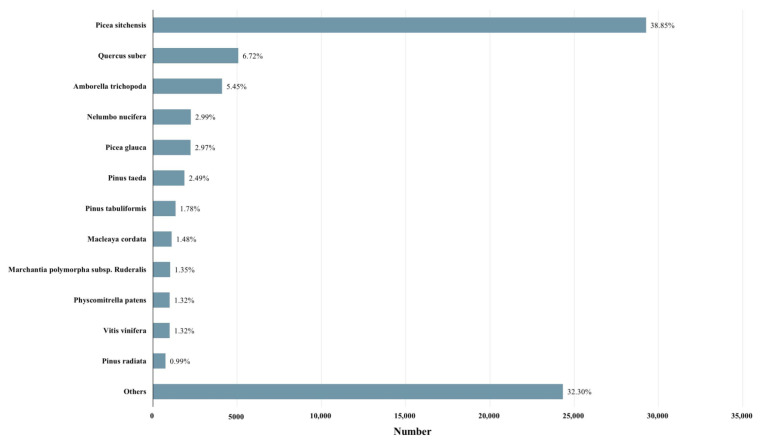
Species distribution of the top BLAST hits for each unigene of *P. koraiensis* under different supplementary lights in Nr.

**Figure 6 ijms-23-13608-f006:**
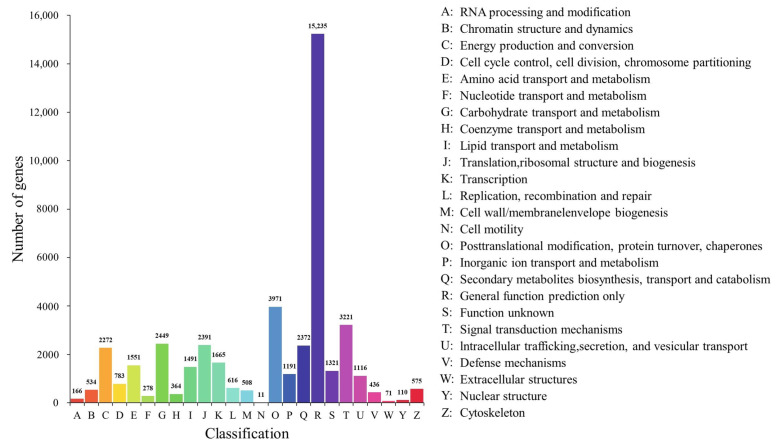
KOG functional classification of assembly of unigenes in *P. koraiensis* under different supplementary lights.

**Figure 7 ijms-23-13608-f007:**
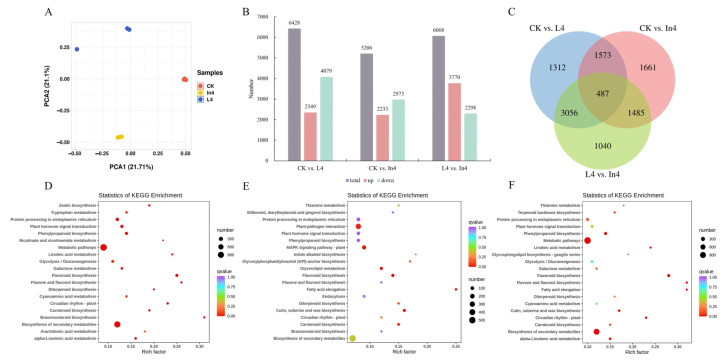
Identification and functional enrichment of differentially expressed genes in *P. koraiensis* under different supplementary lights. (**A**) PCA score plot of expression profiles from different samples. (**B**) The statistics of upregulated and downregulated DEGs in different comparison groups. (**C**) Venn diagram of DEGs in different comparison groups. (**D**) The top 20 KEGG enrichment pathways of DEGs in CK vs. L4. (**E**) The top 20 KEGG enrichment pathways of DEGs in CK vs. In4. (**F**) The top 20 KEGG enrichment pathways of DEGs in L4 vs. In4. CK: Control. L4: Supplementary LED for 4 h. In4: Supplementary incandescent lamp for 4 h.

**Figure 8 ijms-23-13608-f008:**
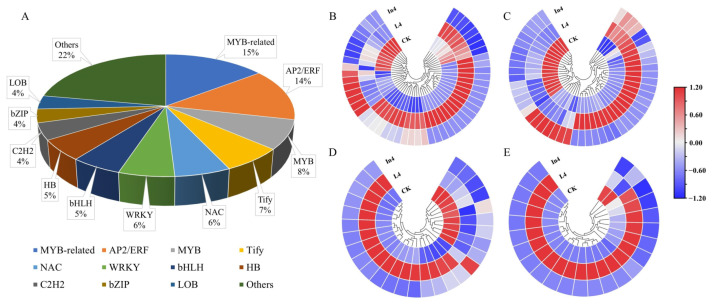
Changes in relative expression levels of DEGs encoding transcription factors in *P. koraiensis* under different supplementary lights. (**A**) The percentage of transcription factor families in all comparison groups. Heatmaps of DEGs involved in (**B**) MYB-related, (**C**) AP2/ERF, (**D**) MYB, and (**E**) Tify transcription factor families. CK: Control. L4: Supplementary LED for 4 h. In4: Supplementary incandescent lamp for 4 h. The color scale from blue to red indicates the expression value from low to high.

**Figure 9 ijms-23-13608-f009:**
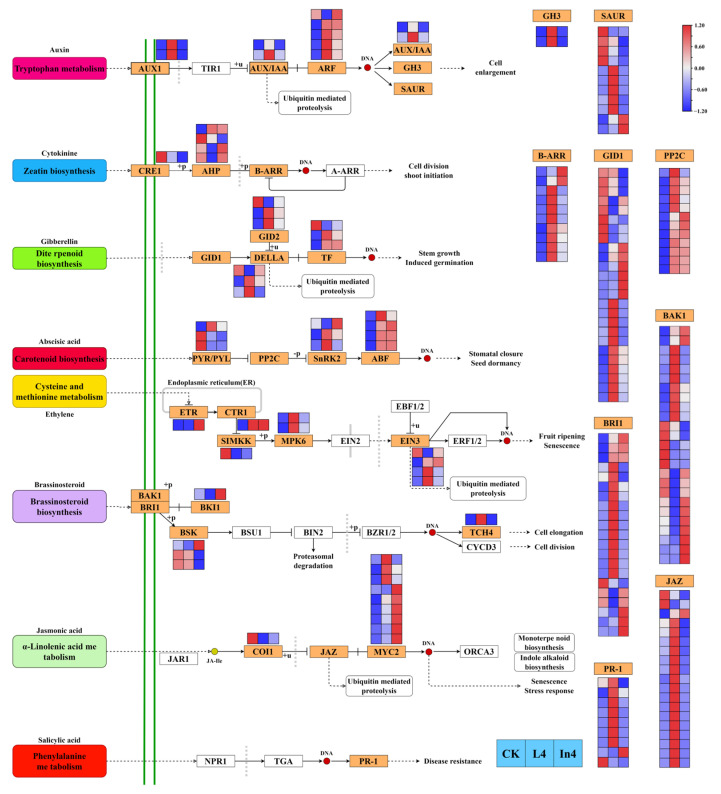
Analysis of DEGs related to phytohormone signaling pathways. The gene expression levels in the three samples are shown in different columns in the colored boxes, and different rows represent different genes. CK: Control. L4: Supplementary LED for 4 h. In4: Supplementary Incandescent lamp for 4 h. The color scale from Min (blue) to Max (red) refers to the expression value from low to high.

**Figure 10 ijms-23-13608-f010:**
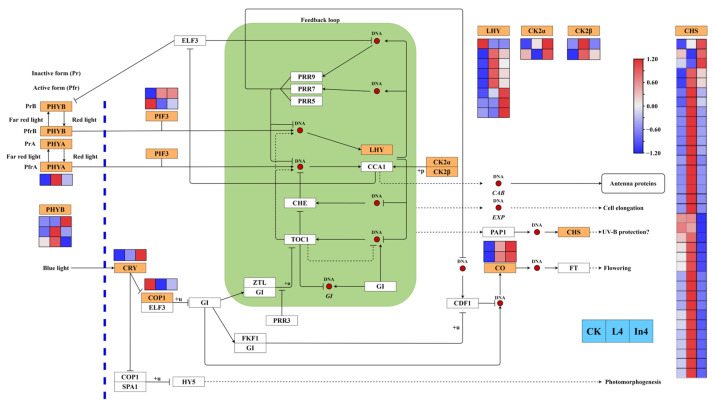
Analysis of DEGs related to circadian rhythm-plant pathway. The gene expression levels in the three samples are shown in different columns in the colored boxes, and different rows represent different genes. CK: Control. L4: Supplementary LED for 4 h. In4: Supplementary Incandescent lamp for 4 h. The color scale from Min (blue) to Max (red) refers to the expression value from low to high.

**Figure 11 ijms-23-13608-f011:**
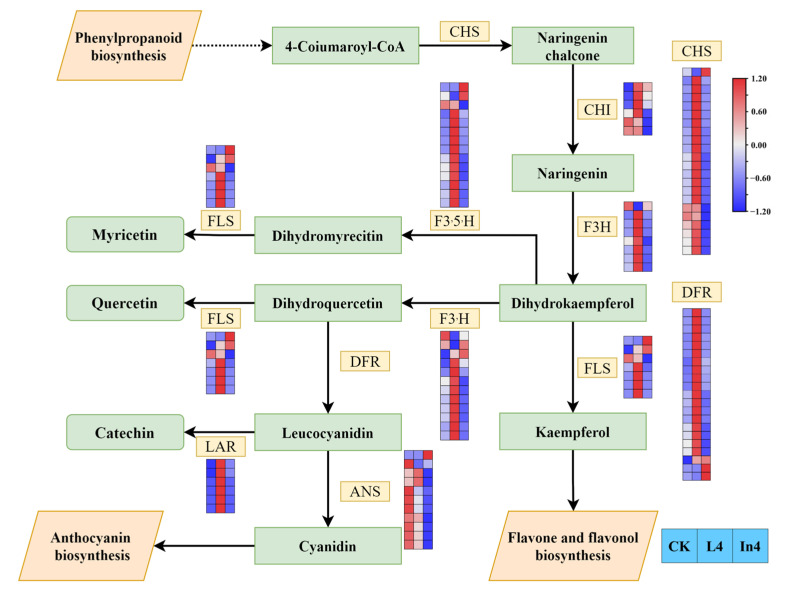
Analysis of DEGs related to flavonoid biosynthesis pathways. The gene expression levels in the three samples are shown in different columns in the colored boxes, and different rows represent different genes. CK: Control. L4: Supplementary LED for 4 h. In4: Supplementary incandescent lamp for 4 h. The color scale from Min (blue) to Max (red) refers to the expression value from low to high.

**Figure 12 ijms-23-13608-f012:**
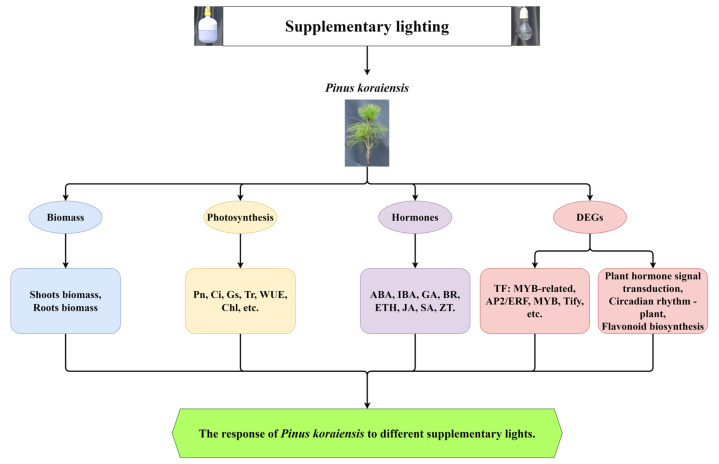
A proposed model of the response of *P. koraiensis* under different supplementary lighting. The model suggested that the biomass, photosynthetic parameters, hormones, TFs, and several metabolic pathways were altered under different supplementary treatments. They all played a role in the response to supplemental lighting.

**Figure 13 ijms-23-13608-f013:**
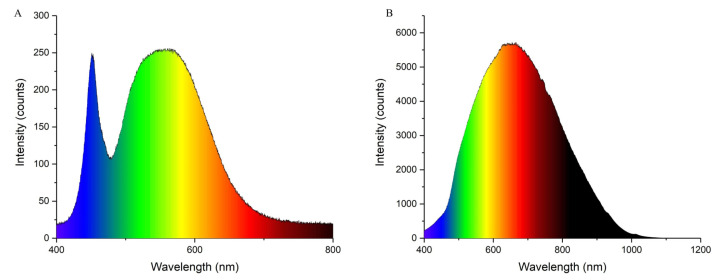
Spectral qualities of the supplementary light sources. (**A**) White LEDs (L) and (**B**) incandescent lamp (In).

## Data Availability

In this study, the data from the three samples were deposited in NCBI under the accession number PRJNA888868.

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
