# Peer review of "Physiological and Transcriptomic Analysis Revealed the Molecular Mechanism of Pinus koraiensis Responses to Light"

_ijms, 2022, doi:10.3390/ijms232113608_

Round 1

Reviewer 1 Report

This paper described some molecular and physiological aspects of Korean pine ‘response to different light spectrums. It seems interesting for particularly ecological studies of that species and pine like gymnosperms. Here some related remarks and suggestions:

-       The figures are difficult to read. The resolution and the size of letters are not good. So all figures from 1 to 5 should be re-prepared.     

-       Line: replace “treatments » with « treatment »

-       The introduction section supplies sufficient background information for the reader to understand and evaluate the experiment the researchers did. Please just mention that pine is belongs to gymnosperms group.

-       Line 110: replace “significantly » with « significant »

-       The paper contains a lot of abbreviations and this can make confusion. So, abbreviations should be defined at first mention in each of the following sections in your paper: title, abstract, text, each figure/table legend. 

-       The figure 2 of changes in plant hormone contents is hard to read. Please use higher size of the figure text. Also the figure 3.

-       Line 134: please change the title to “transcriptomic analysis”

-       Figure 4: please separate this figure on three different figures (Functional annotation, KOG functional classification of assembly unigenes and Similarity)

-       Figure 7 . White text on a yellow and green background provides low value contrast. Please change the text color in this fig.

-       Line 305: this sentence “all the photosynthetic parameters were signifi- 305 cantly decreased in supplementary light compared with the control” seems in contradiction with the abstract sentence said “Supplemental light induces chlorophyll a, chlorophyll b, total chlorophyll and carotenoid accumulation”. Please check or rephrase..

-       The discussion part should be more specific. There is several general information, which is not needed to discuss the results like “…the duration of light in one day is calling photoperiod”. This section should be reformulated to be more related to obtained results.  

-       The genetic aspects of the reproduction of plant material should be clarified. Because if there is intra-population diversity of pine, the transcriptomic analysis will be difficult and need to have genetic identity to avoid any other effects than lights (line 391-396).

Author Response

Thank you for the reviewers’ comments concerning our manuscript entitled “Physiological and transcriptomic analysis revealed molecular mechanism of Pinus koraiensis responses to light”. Those comments are all valuable and very helpful for revising and improving our paper, as well as the important guiding significance to our researches. We have studied comments carefully and have made correction which we hope meet with approval. The main corrections in the paper and the responds to the reviewer’s comments are as flowing:

Point 1: The figures are difficult to read. The resolution and the size of letters are not good. So all figures from 1 to 5 should be re-prepared.

Response: Thanks for your comments. We had made correction according to the comments. Due to picture will be compressed in the word, we have uploaded the sharper figures to the system.

Point 2: Line: replace “treatments » with « treatment ».

Response: Thanks for your comments. We had made correction according to the comments.

Line 18-20: In this study, the one with no supplementary light was used as the control, and two kinds of light sources were set up light-emitting diode (LED) and incandescent lamp, to supplement light treatment of Korean pine.

Line 21-23: The results showed that the growth and physiological-biochemical indicators were significantly different under different supplementary light treatment. The biomass of supplementary light treatment was significantly lower than the control.

Point 3: The introduction section supplies sufficient background information for the reader to understand and evaluate the experiment the researchers did. Please just mention that pine is belongs to gymnosperms group.

Response: Thanks for your comments. We had made correction according to the comments.

Line 37-38: Korean pine (Pinus koraiensis Sieb. et Zucc.) belonging to the Pinus genus and the Pinaceae family, gymnosperms group.

Point 4: Line 110: replace “significantly » with « significant ».

Response: Thanks for your comments. We had made correction according to the comments.

Line 118-119: For photosynthesis analysis, all photosynthetic parameters showed significant differences (p < 0.01) in all treatments (Supplementary Table S1).

Point 5: The paper contains a lot of abbreviations and this can make confusion. So, abbreviations should be defined at first mention in each of the following sections in your paper: title, abstract, text, each figure/table legend.

Response: Thanks for your comments. We had made correction according to the comments.

Point 6: The figure 2 of changes in plant hormone contents is hard to read. Please use higher size of the figure text. Also the figure 3.

Response: Thanks for your comments. We had made correction according to the comments. Due to picture will be compressed in the word, we have uploaded the sharper figures to the system.

Point 7: Line 134: please change the title to “transcriptomic analysis”

Response: Thanks for your comments. We had made correction according to the comments.

Line 154: Transcriptomic analysis

Point 8: Figure 4: please separate this figure on three different figures (Functional annotation, KOG functional classification of assembly unigenes and Similarity)

Response: Thanks for your comments. We had made correction according to the comments. We have separate figure 4 into figure 4, 5 and 6.

Point 9: Figure 7. White text on a yellow and green background provides low value contrast. Please change the text color in this fig.

Response: Thanks for your comments. We had made correction according to the comments. We have uploaded the new figure (figure 9) to the system.

Point 10: Line 305: this sentence “all the photosynthetic parameters were signifi- 305 cantly decreased in supplementary light compared with the control” seems in contradiction with the abstract sentence said “Supplemental light induces chlorophyll a, chlorophyll b, total chlorophyll and carotenoid accumulation”. Please check or rephrase.

Response: Thanks for your comments. We had made correction according to the comments.

Line 390-395: These reduces the net photosynthetic rate of Korean pine under supplementary light condition, while the chlorophyll content increases under supplementary light condition. Plants increase the net photosynthetic rate by increasing chlorophyll content at low the net photosynthetic rate. However, the photoinhibition of Korean pine caused by excessive light causes the net photosynthetic rate to decrease.

Point 11: The discussion part should be more specific. There is several general information, which is not needed to discuss the results like “…the duration of light in one day is calling photoperiod”. This section should be reformulated to be more related to obtained results.

Response: Thanks for your comments. We had made correction according to the comments.

Line 415-416: In this study, supplementary lighting at night extends the duration of light, therefore changes the photoperiod of the Korean pine.

Line 419-420: And CCA1 and LHY are reported in lighting regulation in the plants [52].

Line 424-427: In the circadian systems, the central oscillator consists of three interlocking auto-regulatory transcriptional feedback loops could be known [53]. It is a complex system that requires more in-depth study by researchers.

Line 430-432: The circadian system is a complex, interconnected, and reciprocally regulated network, and the correct timing of circadian rhythms conferred an adaptive advantage [6, 55].

Line 458-460: A large number of studies have demonstrated that flavonoids can respond to changes in light environment.

Point 12: The genetic aspects of the reproduction of plant material should be clarified. Because if there is intra-population diversity of pine, the transcriptomic analysis will be difficult and need to have genetic identity to avoid any other effects than lights (line 391-396).

Response: Thanks for your comments. We had made correction according to the comments. The experimental material in this study is the Korean pine of the same family. All materials are collected using the mixed sampling, which reduced the differences caused by genetic inconsistencies. And the results of the assembly quality showed that 1614 genes were detected with a coverage rate of 91.90% (1482 genes). The average of 85.75% clean reads were mapped to the assembled sequences. These results indicated a high-quality assembly and a high alignment rate.

Reviewer 2 Report

I have reviewed your manuscript and have some concerns regarding manuscript.  They are mentioned throughout the manuscript. Please see the attached comments.

Author Response

Thank you for the reviewers’ comments concerning our manuscript entitled “Physiological and transcriptomic analysis revealed molecular mechanism of Pinus koraiensis responses to light”. Those comments are all valuable and very helpful for revising and improving our paper, as well as the important guiding significance to our researches. We have studied comments carefully and have made correction which we hope meet with approval. The main corrections in the paper and the responds to the reviewer’s comments are as flowing:

Point 1: Authors are suggested to cite more recent articles instead of old references (where applicable).

Zahedi, S.M., Sarikhani, H. The effect of end of day far-red light on regulating flowering of short-day strawberry (Fragaria × ananassa Duch. сv. Paros) in a long-day situation. Russ J Plant Physiol 64, 83–90 (2017). https://doi.org/10.1134/S1021443717010198

Response: Thanks for your comments. We had made correction according to the comments.

Point 2: Line 89: Please check the format.

Response: Thanks for your comments. We had made correction according to the comments.

Point 3: Line 149-152: Too long sentences. Can be rewrite in a better way.

Response: Thanks for your comments. We had made correction according to the comments.

Line 169-171: Gene function annotation was carried out, and the unigene sequences were mapped to public databases by using DIAMOND [32] BLASTX, the results as following. A total of 79,318 (46.76%) unigenes matched to a sequence in at least one of the following databases.

Point 4: Line 291: Perhaps, a model/illustration describing the overall mechanisms of the light in Pinus koraiensis plants can further improve the discussion and summary.

Response: Thanks for your comments. We had made correction according to the comments. We have made a model (figure 12).

Point 5: Material and methods section need to improve. Better to be more descriptive.

Response: Thanks for your comments. We had made correction according to the comments.

Line 565-569:

Where A664, A649 and A470 are measured absorbance at 664, 649 and 470nm, respectively.

Line 571-572: Three seedlings were randomly selected for each treatment, leaves of the same position were collected and mixed, wiped clean and frozen in liquid nitrogen immediately.

Line 587-591: In order to obtain high-quality clean reads, the raw reads were filtered by Fastp software [60]. Firstly, adaptor containing reads were removed; secondly, the reads containing more than 5% ambiguous nucleotides were removed, and finally, the low-quality reads that contained more than 15% bases with q-value ≤ 19 were removed, and the clean reads were obtained for de novo assembly.

Point 6: In the Experimental section, the authors didn’t provide the information that how many biological replicates were used in each treatment to perform one independent experiment. How many plants were used in one biological replicate? How many independent experiments were carried out? Please provide the detail information about the sample size, biological replicates and independents experiments.

Response: Thanks for your comments. We had made correction according to the comments.

Line 503-506: Three P. koraiensis seedlings were randomly selected from each treatment, and the leaves at the same position were selected for mixed sampling, which were immediately frozen in liquid nitrogen to measure the phytohormones. Three biological replicates in each treatment.

Point 7: The authors didn’t provide the information that which leaf/leaves (position on plants), they have used for biochemical analysis.

Response: Thanks for your comments. We had made correction according to the comments.

Line 522-523: Each Korean pine seedling chooses needle leaves of the same height and the same direction to measure.

Line 526-527: Three seedlings were randomly selected from each treatment, the leaves in the same position were selected, wiped clean and mixed.